# *Candida albicans* Mannosidases, Dfg5 and Dcw1, Are Required for Cell Wall Integrity and Pathogenesis

**DOI:** 10.3390/jof10080525

**Published:** 2024-07-27

**Authors:** Maryam Razmi, Jaewon Kim, Jennifer Chinnici, Sujay Busarajan, Hema Vuppalapaty, Deepika Lankipalli, Rui Li, Abhiram Maddi

**Affiliations:** 1Department of Periodontics & Endodontics, School of Dental Medicine, University at Buffalo, Buffalo, NY 14214, USA; mrazmi@buffalo.edu (M.R.);; 2Oral Biology, School of Dental Medicine, University at Buffalo, Buffalo, NY 14214, USA; sujayreddy36@gmail.com (S.B.); hemavupp@buffalo.edu (H.V.); lankipallid@gmail.com (D.L.); 3Department of Restorative Dentistry, School of Dental Medicine, University at Buffalo, Buffalo, NY 14214, USA; rli7@buffalo.edu; 4Division of Regenerative Sciences & Periodontics, Department of Advanced Specialty Sciences, James B. Edwards College of Dental Medicine, Medical University of South Carolina, Charleston, SC 29425, USA

**Keywords:** Dfg5, Dcw1, chitin, cell wall, HOG MAPK, *Candida albicans*, fungi, antifungal drug targets, candidiasis

## Abstract

*Candida albicans* is an oral mucosal commensal fungus that transforms into an opportunistic pathogen under specific conditions, including immunosuppression. It causes oral and systemic candidiasis, which results in a significant health burden. Furthermore, an alarming rise in antifungal drug resistance in *Candida* species raises the urgent need for novel drugs and drug targets. *C. albicans* Dfg5 and Dcw1 are homologous cell wall alpha-1,6-mannosidases with critical functions and represent potential new drug targets. Our past studies have shown that Dfg5 and Dcw1 function in cell wall biogenesis through the cross-linking of glycoproteins into the cell wall, thus playing a key role in cell wall integrity. Additionally, Dfg5 and Dcw1 are required for hyphal morphogenesis. However, the exact functions of Dfg5 and Dcw1 in cell wall integrity, hyphal morphogenesis, and pathogenesis are not known. In this study, we determined the relation of Dfg5 and Dcw1 with Hog1 MAPK, which plays a key role in cell wall integrity via the regulation of chitin synthesis in *C. albicans*. Additionally, we also determined the effects of *dfg5* and *dcw1* mutations on the gene expression of transcriptional regulators of hyphal morphogenesis. Furthermore, we determined the effects of *dfg5* and *dcw1* mutations on pathogenesis in a mouse model of oral candidiasis. Our results demonstrate that *dfg5* and *dcw1* mutations, as well as a *hog1* knockout mutation, result in the dysregulation of chitin synthesis, resulting in a cell separation defect. Heterozygous and conditional mutations in *dfg5* and *dcw1* resulted in decreased transcriptional levels of *cst20*, a positive regulator of hyphal morphogenesis. However, *dfg5* and *dcw1* mutations resulted in increased levels of all the five negative regulators of hyphal morphogenesis—Tup1, Nrg1, Mig1, Rbf1, and Rfg1. Additionally, Tup1 levels were significantly higher than other negative regulators, indicating that Dfg5 and Dcw1 function in hyphal morphogenesis by repressing Tup1. Finally, *dfg5* and *dcw1* mutations affected the ability of *C. albicans* to cause oral candidiasis in mice. Thus, the cell wall glycosidases Dfg5 and Dcw1 are required for virulence and pathogenesis and represent novel drug targets.

## 1. Introduction

*Candida albicans* is a pathogenic fungus that causes oral mucosal, vaginal, and systemic candidiasis in millions worldwide. According to the Centers for Disease Control (CDC), 5–7% of infants <1 month of age develop oral candidiasis, while the prevalence among AIDS patients is estimated to range from 9 to 31% [1]. Furthermore, the clinical incidence of oral candidiasis is nearly 90% in cancer patients, with *Candida albicans* isolated from 58% of patients [2]. The risk of systemic/invasive candidiasis is also increased in immunocompromised patients and may become life-threatening. Hence, the diagnosis and treatment of oral/mucosal candidiasis is critical. However, there are numerous reports of *Candida* species becoming resistant to the currently available antifungal agents [1,2]. As a result, there is a critical need for drug targets to develop novel antifungal drugs. 

The *C. albicans* cell wall is an ideal target for drugs due to its extracellular location and accessibility [3]. Furthermore, it is the only organelle that is present in the fungal cell and absent in the host. The cell wall of *C. albicans* is a complex structure made of polysaccharides and cell wall proteins [3]. The polysaccharides form a three-dimensional matrix of branched β-1,3-glucan, to which β-1,6-glucan and chitin are attached through their reducing ends [3]. Chitin has important functions in cell wall integrity and morphogenesis, as well as septum formation during cell division [4]. *C. albicans* has four chitin synthases—Chs1, Chs2, Chs3, and Chs8 [4]. Chs1 is a class II chitin synthase and is the only chitin synthase required for cell viability [5,6]. Chs1 is required for primary-septum formation and also lateral wall formation, thus affecting cell wall integrity [5,6]. Chs3 is a class IV chitin synthase and is responsible for the synthesis of the majority of the chitin present in the cell walls of yeast and hyphae [7]. Chs2 and Chs8 are both class I chitin synthases [4]. These chitin synthases work in a coordinated fashion in order to synthesize chitin in the cell wall and septum. Chs8 is known to synthesize long-chitin microfibrils and Chs3 short-chitin rodlets of chitin [4]. Studies of individual mutants of these chitin synthases in *C. albicans* indicate that they have compensatory functions [4].

A majority of cell wall proteins are covalently cross-linked into the polysaccharide matrix by glycosidase enzymes present in the cell wall space [3,8]. *C. albicans DFG5* and *DCW1* encode for glycosidase/mannanase/mannosyltransferase enzymes (*gh-76* family) that are targeted to the cell wall space by the N-terminal signal for secretion and a C-terminal GPI anchor [9]. Past studies in *Saccharomyces cerevisiae* have shown that the *dfg5*/*dcw1* knockout mutation is lethal, indicating that these cell wall proteins fulfill essential cellular functions [9,10]. Our study in *Neurospora crassa* demonstrated that these proteins function in cell wall protein incorporation into the wall and thus affect cell wall biogenesis [11]. These studies clearly indicate that *DFG5* and *DCW1* have highly conserved functions and play an important role in fungal cell physiology. In *C. albicans*, *dfg5* and *dcw1* single mutants are viable; however, the *dfg5*/*dcw1* knockout mutant is lethal, indicating a functional redundancy [12]. In addition, Dfg5 has been shown to be in the cell membrane, and the expression of *HWP1*, a hypha-specific gene, is affected in the *dfg5* knockout mutant [12]. Such alterations in specific gene expression occur when signal transduction pathways are affected. It was confirmed in *Saccharomyces cerevisiae* that Hog1 and Slt2 cell signaling pathways are affected in the *dfg5*Δ mutant [13]. Further evidence in the mycoparasite *Trichoderma atroviride* also suggests that Dfg5 plays a critical role in hyphal morphogenesis and osmoregulation via MAPK signaling [14].

Our past studies in *C. albicans* have utilized the *pMET3*-modulated *dfg5*/*dcw1* conditional mutant described previously [12]. We showed that *DFG5* and *DCW1* function in the covalent incorporation of cell wall proteins and thus play critical roles in cell wall biogenesis [15]. Our data also indicate that *dfg5*/*dcw1* mutants are affected in hyphal morphogenesis and biofilm formation. Additionally, we also showed that basal Hog1 MAPK levels are reduced in the *dfg5*/*dcw1* mutants and that *dfg5*/*dcw1* mutants have a cell separation phenotype similar to the *hog1* knockout mutant [16]. Hog1 MAPK, in turn, is known to regulate chitin synthesis by controlling the expression of chitin synthase genes in *C. albicans* [17]. However, the exact functions of Dfg5 and Dcw1 in the regulation of chitin synthesis and hyphal morphogenesis remain unknown. Thus, the objective of this study was to determine whether Dfg5 and Dcw1 affect the transcriptional regulation of chitin synthesis and hyphal morphogenesis, key physiological processes for the virulence and pathogenesis of the opportunistic pathogen *Candida albicans*. 

## 2. Materials and Methods

### 2.1. Strains and Growth Conditions

The genetic backgrounds of the *Candida albicans* strains used in this study can be found in Table 1. The strains were cultured in Yeast Nitrogen Base (YNB) medium with ammonium sulfate and 2% glucose. A synthetic complete supplement mixture (MP Biomedicals, Santa Ana, CA, USA) was added as an amino acid supplement to YNB. In addition, 5 mM methionine and 2 mM cysteine were added to the medium for the ES195 strain for the conditional repression (85%) of the chimeric *MET3::DFG5* gene to generate a Dfg5p-deficient condition. The strains have been described previously [12] and were kindly donated by Dr. Aaron Mitchell (Augusta University, Augusta, GA, USA) or obtained from FGSC (Fungal Genetics Stock Center, Kansas State University, Manhattan, KS, USA).

### 2.2. Light and Fluorescence Microscopy Analysis

Overnight cultures were diluted to an OD_600_ of 1.5 (approximately 5 × 10^7^ CFU/mL) in a total volume of 1 mL of YNB either with or without 1.668 µg/mL chitinase (Sigma, Burlington, MA, USA), as described previously [18]. The ES195 strain was grown with and without methionine and cysteine for control cultures and chitinase-treated cultures. Cultures were allowed to incubate at room temperature with shaking at 225 rpm for 3 h. Cells were pelleted by centrifugation at 900× *g* for 2 min. The culture medium was removed, and the cells were resuspended in 1× PBS containing 100 µg/mL Calcofluor White [19,20]. Then, 3 µL of each sample was immediately placed on a microscope slide with a coverslip and imaged with a Nikon Eclipse TE2000-U (Nikon, Konan, Tokyo, Japan) at 400× total magnification using Spot Advanced 4.0.4 software. Fluorescence microscopy of CFW was performed using a UV filter. False color was added to the fluorescent images with ImageJ software (Version 1.x). The calcofluor white (CFW) fluorescence intensity of 50 cells/strain was calculated for two separate experiments to obtain a total of 100 cells/strain, and the background directly next to each of these cells was measured using ImageJ software, as described previously [19]. Corrected Total Cell Fluorescence (CTCF) calculations were performed to quantify chitin accumulation in each strain as follows: CTCF = Integrated density − [(Area of selected cell) × (Mean fluorescence of background readings)].

### 2.3. Scanning Electron Microscopy (SEM) Analysis

SEM analysis was performed as described previously [21]. The cultures were prepared for light microscopy and then transferred to 6-well polystyrene plates, where the cells were allowed to settle for 90 min at 37 °C on Fetal Bovine Serum (FBS, Seradigm)-coated glass squares. The cells were fixed and dried. The samples were coated with evaporated carbon at high vacuum (Denton 502 Evaporator). SEM images were acquired with a Hitachi SU70 FESEM (Hitachi, Tokyo, Japan) at 2.0 KeV using the lower detector and no tilt. SEM images were analyzed from two separate experiments.

### 2.4. Quantitative Real Time PCR (qRT-PCR) Analysis

RT-qPCR analysis was performed as described previously [22]. Primers were prepared for *CHS* genes (*CHS1*, *CHS2*, *CHS3*, and *CHS8*), positive hyphal transcriptional regulators (*CST20*, *HST7*, *CPH1*, and *CPH2*), negative hyphal regulators (*TUP1*, *NRG1*, *RBF1*, *RFG1*, and *MIG1*), and the housekeeping gene *EFB1* (Table 2). The RT-qPCR reactions were performed using the Applied Biosystems 7500 Real Time PCR machine (Applied Biosystems, Waltham, MA, USA) with the standard cycling protocol from the SYBR Green FastMix product manual: denaturation at 95 °C for 1 min, annealing for 40 cycles of 58–64 °C for 5 s, and extension at 60 °C for 34 s. Data were collected at the end of the extension step. To analyze the data, the values for the *CHS* genes were normalized to the *EFB1* housekeeping gene for the 2^−ΔΔCt^ calculations using Microsoft Excel (Version 2023, Microsoft, Seattle, WA, USA). RT-qPCR experiments were performed for two separate experiments with triplicates of each strain/sample. The standard deviation values are for a minimum of 6 samples per strain. For statistical analysis of significance, Student’s *t*-tests with equal variances were performed with a *p*-value < 0.05. 

### 2.5. Mouse Model Protocol for Oral Candidiasis 

Oral candidiasis infections were established in mice as previously described using a protocol approved by the University at Buffalo IACUC (Protocol #201700003) [25]. Five BALB/c mice (11-week-old male and female mice) (Jackson Labs) were infected for each strain used (5 groups of mice in total) (Table 1). Groups infected with SC5314 (group 1) and DAY185 (group 2) were used as controls. These were compared with groups infected with the mutant strain ES1 (group 3) and the conditional mutant ES195. For the ES195 strain, one group of mice was infected with untreated cells (group 4), and one group of mice was infected with cells that had been pre-treated with 5 mM methionine and 2 mM cysteine (Bulksupplements.com accessed on 04.01.2019) for one hour prior to infection (group 5) to achieve 85% repression of the remaining copy of *DFG5* in this strain. To maintain this repression throughout the experiment, group 5 mice received methionine and cysteine in their drinking water. Immunosuppression was induced by administering 225 mg/kg cortisone 21-acetate (Sigma Aldrich, Burlington, MA, USA) subcutaneously on the day prior to infection and on days 1 and 3 post-infection (Figure 1). The infected mice were monitored for changes in behavior and health. Pictures of the tongue were taken on days 1, 3, and 5 post-infection. On day 5, mice were euthanized by cervical dislocation performed under anesthesia (Ketamine/Xylazine as described above). The tongues and surrounding hypoglossal tissue were removed and cut in half lengthwise. One half was used for histopathological analysis by H&E (Hematoxylin and Eosin) staining as well as PAS (Period Acid Schiff) staining after fixation with 10% Neutral Buffered Formalin (IMEB). The other half was weighed and homogenized completely for the quantification of infection by colony forming unit (CFU) assessment. For CFU experiments, samples were plated in triplicate for each sample. 

### 2.6. Statistical Analysis

Statistical analysis was performed using Microsoft Excel on a Windows operating system. Each experimental group had a triplicate of samples. For CTCF calculations, the five experimental groups were compared using paired *t*-tests assuming unequal variances. For RT-qPCR, the experiments were always performed in triplicate for each group and repeated twice. For RT-qPCR, the experimental groups were compared using Student’s *t*-test for two samples assuming equal variances. For CFU analysis, homogenized tongue samples were plated in triplicate for each sample and analyzed by Student’s *t*-test for two samples assuming equal variances. A *p*-value of <0.05 was considered significant for all experiments.

## 3. Results

### 3.1. DFG5 and DCW1 Mutations Result in a Cell Separation Defect Identical to the hog1 Mutant, Confirmed by Light, Fluorescence, and Electron Microscopy

Our past studies showed that mutants of *DFG5*/*DCW1* have reduced basal Hog1 levels [16]. In order to compare and contrast the phenotypes of *DFG5*/*DCW1* mutants with the *hog1* mutant, we performed light microscopy, fluorescence microscopy, and SEM. Light microscopy analysis of ES195 and ES195+M/C conditional mutant strains indicated that they have a cell separation defect, as compared to control strains (Figure 2). This cell separation defect was also confirmed for the *hog1* knockout mutant (Figure 2A). Fluorescence imaging using CFW, which binds to chitin, has been used for measuring chitin levels in *C. albicans* [19,20]. CFW fluorescence analysis revealed a higher intensity of fluorescence at the cell septae, indicating increased chitin accumulation (Figure 2A). CTCF measurements, which corresponded to chitin accumulation, were performed for 100 cells/strain. There was significantly higher CTCF for ES195+M/C and Hog1 as compared to the WT (SC5314) strain (Figure 2C). Additionally, significantly higher CTCF was observed for ES195+M/C as compared to Hog1. There was no significant difference in CTCF values for ES1 as compared to the WT. It is interesting that this cell separation defect and even the increased intensity of CFW fluorescence were only minimal for the ES1 mutant, in which both copies of *DFG5* are mutated and only one functional copy of *DCW1* is present. This may indicate that the one remaining copy of *DCW1* is sufficient to compensate for the loss of both copies of *DFG5.* CFW fluorescence was significantly higher for ES195+M/C as compared to other strains, indicating a higher accumulation of chitin levels in the conditional mutant. Our data indicate that the suppression of *DFG5*/*DCW1* results in an increase in chitin levels in the cell wall and may lead to the cell separation defect.

### 3.2. Chitinase Treatment Results in Reversal of Cell Separation Phenotype for dfg5/dcw1 Mutants

We then wanted to determine whether the cell separation defect occurs as a result of increased chitin accumulation. A characteristic of the cell separation defect in the *hog1* knockout mutant is its reversal by treatment with commercially available chitinase [18]. Mutant and control strains were incubated with chitinase for 3 h, which resulted in the improvement of cell separation for the ES195, ES195+M/C conditional mutant, and *hog1* knockout mutant strains, indicating that the identical phenotype among the mutant strains was due to the abnormal increase in chitin accumulation (Figure 2A). This was further confirmed by SEM analysis, which showed that the cell separation defect was due to the lack of separation of the mother–bud neck following cell division (Figure 2B).

### 3.3. Dfg5 and Dcw1 Affect Gene Expression of Chitin Synthases CHS1, CHS2, CHS3, and CHS8

We then wanted to determine whether the increased chitin levels in DFG5/DCW1 mutants occurred as a result of the increased expression of chitin synthase (*CHS*) genes in *C. albicans*. Transcriptional analysis of *CHS* genes was performed for control and mutant strains under basal and chitin stress (CFW) conditions using RT-qPCR analysis (Figure 3). Analysis of the control strains—WT and DAY185—indicated that the level of expression of all four chitin synthases, *CHS1*, *CHS2*, *CHS3*, and *CHS8*, was almost identical, indicating that, under normal conditions, these chitin synthases may be produced in similar quantities. However, *dfg5*/*dcw1* heterozygous mutations resulted in variable regulation of chitin synthase gene expression. Under basal conditions, the ES1 mutant was unable to upregulate the expression of any of the four chitin synthases at either the 6 h or 12 h time point. This may indicate that Dfg5 is required for the upregulation of all four chitin synthases. This could be important under cell wall stress conditions, considering that the ES1 mutant is already under cell wall stress due to its inability to cross-link proteins in the cell wall. The ES195 mutant only had a modest increase (2–3-fold) in the expression of all four chitin synthase genes at the 6 h time point, and their expression further increased (3–5-fold) at the 12 h time point. This result indicates that one copy of *DFG5* is sufficient to compensate for the lack of *DCW1* and thus can upregulate the chitin synthases modestly. Only a modest increase in expression may occur in the ES195 mutant due to a possible modest activation of the PKC pathway occurring in response to cell wall stress [17]. On the other hand, the ES195+M/C conditional mutant showed an increase in the gene expression of *CHS2* (6–7-fold), *CHS3* (10–11-fold), and *CHS8* (3–4-fold) at the 6 h time point. However, at the 12 h time point, this gene expression further increased for *CHS2* (10–12-fold), *CHS3* (12–14-fold), and *CHS8* (6–8-fold). On the other hand, *CHS1* gene expression remained low. This information indicates that very low levels of Dfg5 (15%) trigger a compensatory upregulation of *CHS2*, *CHS3*, and *CHS8* while ignoring *CHS1*. 

### 3.4. DFG5/DCW1 Conditional Knockout Mutations Result in Decreased Levels of Cst20, a Positive Transcriptional Regulator of Hyphal Morphogenesis

Hyphal morphogenesis in *C. albicans* is regulated by several positive and negative transcriptional regulators [26,27]. Positive transcriptional regulators promote the transition from yeast to the hyphal form. Although it is known that Dfg5 and Dcw1 are required for hyphal morphogenesis in vitro, whether *dfg5*/*dcw1* heterozygous mutations affect transcriptional regulators of hyphal morphogenesis is not known. Thus, we performed RT-qPCR analysis to determine whether DFG5/DCW1 mutants are affected in their expression of the positive transcriptional regulators Cst20, Hst7, Cph1, and Cph2. In Figure 4A, the gene expression analysis of positive transcriptional regulators under basal conditions (30 °C) is shown. The most significant increase in gene expression was observed for the *CST20* transcriptional regulator for the ES195 and ES195+M/C strains. Figure 4B demonstrates gene expression analysis under hyphal-inducing conditions (Spider medium). In this case, there is likewise increased gene expression with respect to the *CST20* transcriptional regulator for the ES195 and ES195+M/C mutants, but this difference is not nearly as pronounced as that shown under basal conditions (Figure 4A). It also depicts gene expression increases for *HST7* and *CPH1* for the ES195 and ES195+M/C strains but lower expression of *CPH2*. When a relative gene expression analysis was performed between the hyphal-inducing and basal conditions (Figure 4C), a significant decrease in gene expression was found for *CST20* in the ES195+M/C conditional mutant strain as compared to the WT. This result indicates that Dfg5 and Dcw1 may function in hyphal morphogenesis by increasing *CST20* levels.

### 3.5. DFG5 /DCW1 Conditional Knockout Mutations Result in Increased Levels of TUP1, RBF1, MIG1, RFG1, and NRG1, Negative Transcriptional Regulators of Hyphal Morphogenesis

The impact of *C. albicans DFG5/DCW1* mutations on several negative transcriptional regulators (*TUP1*, *RBF1*, *RFG1*, *MIG1*, *NRG1*) was also determined (Figure 4D). The gene expression of all negative transcriptional regulators was significantly upregulated in the *DFG5*/*DCW1* conditional mutant, ES195, and ES195+M/C strains as compared to ES1 and the WT. Tup1 is a major hyphal regulator and serves as a transcriptional suppressor [26,27]. Of note here is the significantly increased levels of expression of Tup1 in both the ES195 and ES195+M/C strains, thus leading to constitutive hyphal expression. Thus, our data show that the suppression of Dfg5 and Dcw1 can lead to the activation of Tup1, resulting in hyphal repression, indicating a putative mechanism for the regulation of hyphal morphogenesis by Dfg5 and Dcw1.

### 3.6. DFG5/DCW1 Mutations Cause Defects in the In Vivo Virulence and Pathogenesis of C. albicans in a Mouse Model of Oral Candidiasis

Although *in vitro* data suggest critical functions for Dfg5 and Dcw1 in hyphal morphogenesis, it is not known whether *DFG5*/*DCW1* mutations have any effect on the virulence and pathogenesis of *C. albicans* in a mouse model of infection. Hence, we performed experiments to determine the in vivo pathogenesis of the wild-type and *DFG5*/*DCW1* mutant strains in a mouse model of oral candidiasis (Figure 5). Mice infected with the WT strain SC5314 and the parental control DAY185 showed oral candidiasis on the tongue. Among the heterozygous mutants, ES1 was able to cause infection. However, the ES195 strain and ES195+M/C strain did not show any visible plaque formation. Figure 5B demonstrates the histological analysis of mouse tongues in the various infection groups. The wild-type strain SC5314 and the DAY185 strain both show that there is destruction of the epithelium and the clear presence of hyphal structures penetrating the tissues. In the ES1 mutant, pseudo-hyphal-type structures were noted, but real hyphae were not observed. Minor destruction of the epithelium was also observed. In the ES195 and ES195+M/C mutant groups, there was no colonization observed, and the epithelia were largely intact, thus indicating that no infection had occurred. Further quantification of *Candida* CFU within the tongue tissue (Figure 5C) indicated that the ES195 strain had reduced CFU, whereas the ES195+M/C strain did not have any (Figure 5C). This demonstrates that cells may be present and survive on the tongue in ES1 and ES195 strains; however, due to a lack of hyphal structure, they are not able to penetrate tissues and ultimately cause disease. However, for the ES195+M/C conditional mutant, the cells could not survive and cause infection. Thus, our data indicate that Dfg5 and Dcw1 are required for virulence and pathogenesis in a mouse model.

## 4. Discussion

The composition and structural organization of the *Candida albicans* cell wall is dynamically regulated in response to changing environmental conditions [3,8]. The carbohydrates chitin and beta-glucan form the structural framework of the fungal cell wall [3]. The alteration and reconstitution of chitin and beta-glucan within the cell wall occur in response to the disruption of genes in the cell wall biosynthetic pathways of *C. albicans* [28]. Antifungal drug resistance in *Candida* species has become a major concern in healthcare. Caspofungin, a drug that belongs to the echinocandin class, is one of the commonly used antifungal agents for treating invasive candidiasis [20]. The upregulation of cell wall chitin levels has been identified as an alternate drug resistance mechanism against caspofungin, independent of mutations in the FKS region [29]. Furthermore, large amounts of chitin in the cell wall correspond to increasing caspofungin resistance in animal models [20,29,30]. *C. albicans* stimulates chitin synthesis to enable cells to survive lethal concentrations of echinocandins, including caspofungin. Similar observations in response to caspofungin were made for multi-drug-resistant *Candida auris* [31]. *C. albicans* has four chitin synthases—Chs1, Chs2, Chs3, and Chs8—that play important roles in cell wall formation and septum formation and affect cell wall integrity [4,6,7]. The transcriptional regulation of chitin synthases in *C. albicans* is controlled by three signaling pathways in a coordinated manner—the Ca^2+^ calcineurin pathway, HOG pathway, and MKC pathway [4]. Recent work on *C. albicans* and its deletion mutants of the β-1,6-glucan synthesis genes *KRE6* and *SKN1* found that cell wall chitin levels increased through the post-transcriptional regulation of the chitin synthase Chs3, leading to cell viability maintenance via Ca^2+^/calcineurin and PKC signaling pathways [32]. Using the *lacZ* reporter assay, it was found that the *hog1∆* mutant had an altered expression of the chitin synthases *CHS3* and *CHS8* in *C. albicans* [25]. In our past studies, we have shown that *C. albicans* Dfg5 and Dcw1 are required for cell wall integrity and have reduced basal Hog1 levels [16]. Thus, it was reasonable to investigate the functions of Dfg5 and Dcw1 in chitin synthesis, a critical component of antifungal drug resistance.

In this study, we compared the *dfg5*/*dcw1* heterozygous mutant to the *hog1* knockout mutant in relation to chitin synthesis. At the 6 h time point, the levels of expression in the *hog1* knockout mutant appeared to be at the WT level. However, at the 12 h time point, the *hog1* knockout mutant had increased expression of the *CHS1*, *CHS2*, *CHS3*, and *CHS8* genes. A plausible reason for this could be that by 12 h, which represents the mid-log phase, the glucose present in the culture medium was depleted, resulting in reduced beta-glucan synthesis and thus weakening the wall. This, in turn, could have triggered alternate pathways, i.e., the PKC pathway and/or the calcineurin pathway, for chitin synthesis. Additionally, *CHS3* expression was not increased in the *hog1* knockout mutant even at the 12 h time point, indicating that Hog1 may be required for its upregulation. Our data are different from those in the past study by Lenardon et al., 2007 [4], in two ways—the methods used and the time points of gene expression measurement. The study by Lenardon et al., 2007 [4], used a promoter-based beta-galactosidase assay to measure gene expression in the presence of CFW, as compared to RT-qPCR under basal and CFW conditions in our study. Also, the beta-galactosidase assay was performed when the cells reached an OD of 1, which would be past the mid-log phase and may represent a different time point than the 12 h in our study. Furthermore, our study also indicates that the gene expression of chitin synthases varies, depending upon the time of growth. 

Our study also determined the functions of Dfg5 and Dcw1 in the hyphal morphogenesis and in vivo pathogenesis of *C. albicans.* Our data indicate that Dfg5 and Dcw1 are required for the increased expression of Cst20, a positive transcriptional regulator of hyphal morphogenesis during hyphal induction. However, the most striking data were related to the significantly higher expression of negative transcriptional regulators (*TUP1*, *RBF1*, *RFG1*, *MIG1*, *NRG1*) of hyphal morphogenesis in the *DFG5*/*DCW1* conditional knockout mutant. These data indicate that Dfg5 and Dcw1 are required for the repression of negative transcriptional regulators, including *TUP1*, which acts as a co-factor for the others (*RBF1*, *RFG1*, *MIG1*). It is noteworthy that there are no known upstream signaling pathways that have been identified for Tup1. This is the first study that describes potential novel upstream functions of Dfg5 and Dcw1 in the negative transcriptional regulation of hyphal morphogenesis. Further, our animal study experiments indicate that Dfg5 and Dcw1 are required for pathogenesis in a mouse model of oral candidiasis. The heterozygous mutant, ES1, is able to cause disease and forms pseudohyphae, as depicted in the histological sections of the tongue. However, the conditional mutants ES195 and ES195+M/C are unable to form hyphal structures or cause disease. 

## 5. Summary and Conclusions

Overall, our data indicate that Dfg5 and Dcw1 cell wall glycosidases regulate cell wall chitin levels by affecting the gene expression of chitin synthases. Furthermore, this phenomenon was similar to that in the *hog1* knockout mutant but more severe, especially as observed for the *dfg5*/*dcw1* conditional mutant. Further, the hyphal morphogenesis pathways also appear to be affected by Dfg5 by affecting Cst20, Tup1, Rbf1, Rfg1, Mig1, and Nrg1. Based on our data, we hypothesize that Dfg5 and Dcw1 act as cell wall sensors and interact with signaling proteins (Sln1, Sho1, and Opy2) within the cell wall that regulate the aforementioned pathways (Figure 6). 

## Figures and Tables

**Figure 1 jof-10-00525-f001:**
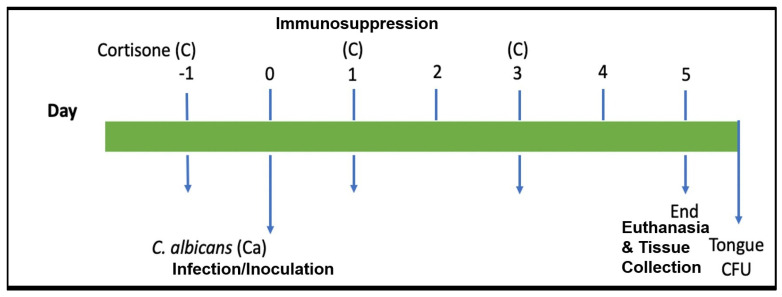
Timeline of mouse infection by *Candida albicans*. Cortisone 21-acetate (225 mg/kg) was given subcutaneously on the day prior to infection (day −1) and days 1 and 3 post-infection to induce immunosuppression. Inoculation was performed on day 0 with 10^6^ cells/mL of *C. albicans* control and mutant strains using a calcium alginate oral swab for 75 min. Mice from all groups were euthanized on day 5 to collect tongue tissue for CFU (colony forming units) and histological analysis.

**Figure 2 jof-10-00525-f002:**
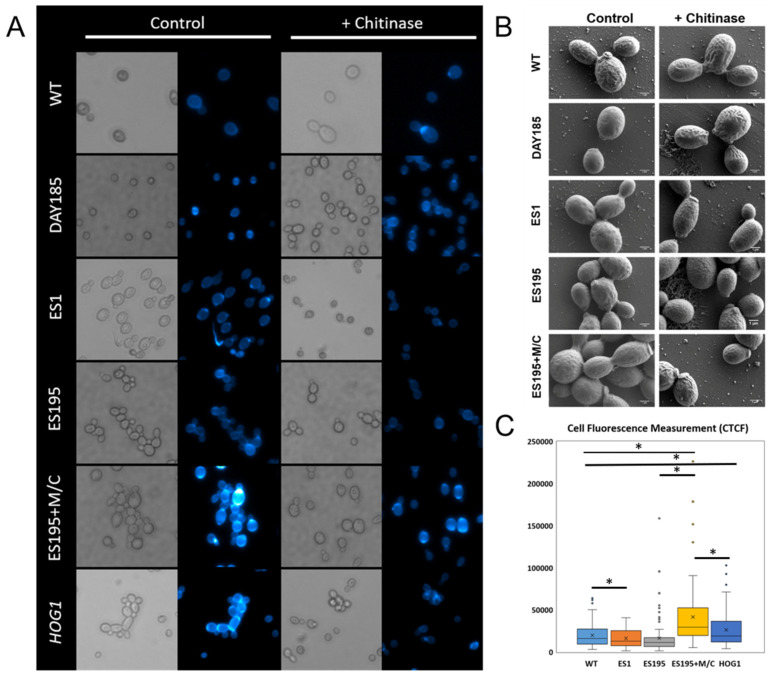
Dfg5 and Dcw1 affect physiological chitin synthesis. (**A**) Light microscopy and fluorescence microscopy analyses of strains were performed using CFW, which binds to chitin. Fluorescence imaging revealed a higher intensity of fluorescence at the cell septae. (**B**) Scanning electron microscopic (SEM) analysis of control and mutant strains treated with chitinase for 3 h was performed. Strain ES195 and the ES195+M/C conditional mutant strain exhibit a cell separation defect, in which the cells remain clumped and are unable to separate following cytokinesis, as compared to the control strains, the WT and DAY185 strains. (**C**) Corrected Total Cell Fluorescence (CTCF) calculations were performed to quantify chitin accumulation in each strain for 100 cells/strain. Statistical analysis was performed using paired *t*-tests between strains; * indicates statistical significance (*p* < 0.05).

**Figure 3 jof-10-00525-f003:**
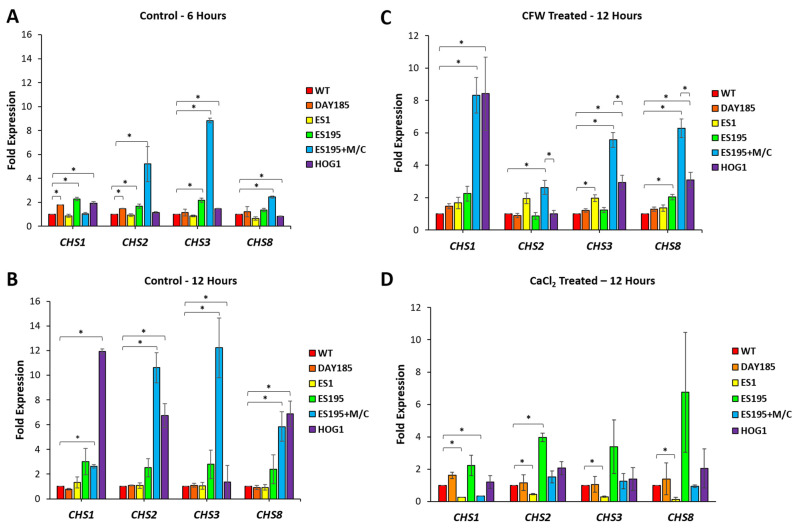
*Candida albicans* Dfg5 and Dcw1 affect chitin synthase gene expression. Transcriptional analysis of *CHS* genes was performed for control and mutant strains under basal conditions at 6 h (**A**) and 12 h (**B**) time points and at the 12 h time point under CFW (**C**) and CaCl_2_ (**D**) conditions using RT-qPCR analysis. Statistical analysis was performed using *t*-tests; * indicates statistical significance as compared to WT (*p* < 0.05).

**Figure 4 jof-10-00525-f004:**
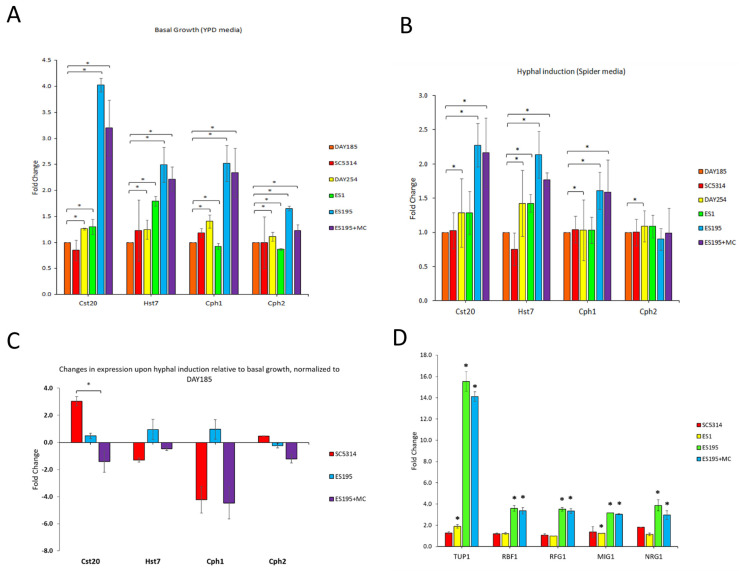
*Candida albicans DFG5* and *DCW1* mutations affect the gene expression of transcriptional regulators of hyphal morphogenesis. (**A**) Gene expression analysis of positive transcriptional regulators under basal conditions (YPD, 30 °C). (**B**) Gene expression analysis of positive transcriptional regulators under hyphal-inducing conditions (Spider medium, 37 °C). (**C**) Relative gene expression analysis between hyphal-inducing and basal conditions indicates that Cst20 is significantly reduced in the conditional mutant as compared to the WT. (**D**) The gene expression of negative transcriptional regulators (Tup1, Rbf1, Mig1, Nrg1, Rfg1) was analyzed under non-inducing or basal conditions (YPD, 30 °C) only. The gene expression of all negative regulators is significantly increased in the conditional mutants as compared to controls. Statistical analysis was performed using *t*-tests; * indicates statistical significance as compared to WT (*p* < 0.05).

**Figure 5 jof-10-00525-f005:**
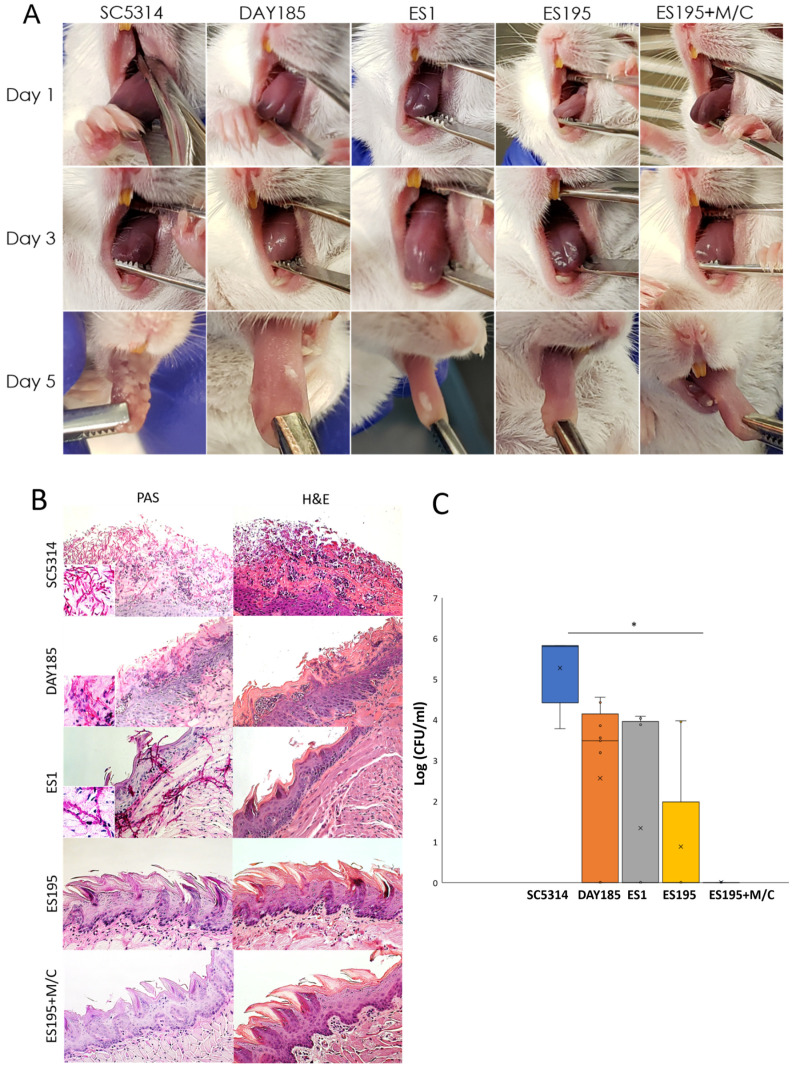
*Candida albicans* Dfg5 and Dcw1 are required for the in vivo pathogenesis of oral candidiasis. Inoculation was performed on day 0 with 10^5^ cells/mL of *C. albicans* control and mutant strains using a calcium alginate oral swab for 75 min. (**A**) Mice infected with the WT strain developed oral candidiasis similarly to those infected with the DAY185 strain. Among the heterozygous mutants, the ES1 strain was able to cause oral infection (as depicted by the presence of tongue plaque), but the ES195 strain and ES195+M/C strain were unable to cause obvious oral candidiasis. (**B**) Histological analysis of the infected tongues of mice was performed. The wild-type strain SC5314 and DAY185 strain both showed infection with oral candidiasis, active colonization, destruction of the epithelium, and the clear presence of hyphal structures penetrating the tissues. (**C**) CFU counts of infected tongues were analyzed for each group of mice infected with a specific strain. CFU counts were significantly lower for the ES195 strain and ES195+M/C conditional mutant strain as compared to the control strains. Statistical analysis was performed using *t*-tests; * indicates statistical significance as compared to WT (*p* < 0.05).

**Figure 6 jof-10-00525-f006:**
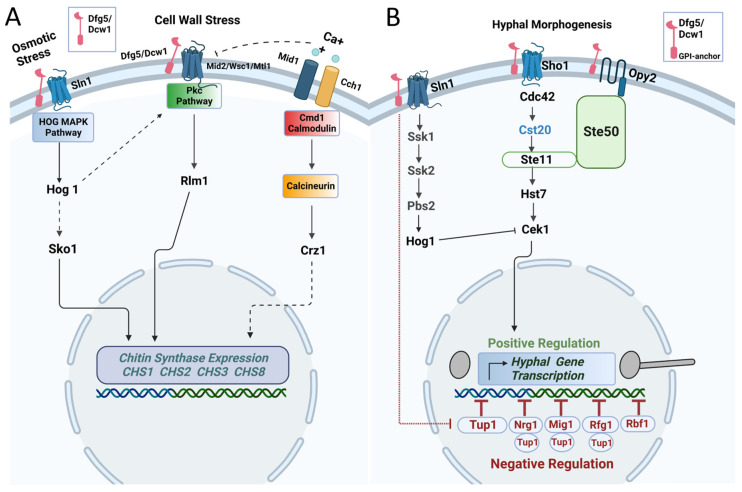
Dfg5 and Dcw1 control chitin synthesis and hyphal morphogenesis. (**A**) Dfg5 and Dcw1 may affect chitin synthesis under basal and cell wall stress conditions by interacting with upstream signaling proteins of MAPK pathways. (**B**) Dfg5 and Dcw1 may affect hyphal morphogenesis by affecting MAPK pathways that regulate transcriptional factors of hyphal morphogenesis. The graphical image was prepared using Biorender.

**Table 1 jof-10-00525-t001:** *Candida albicans* strains and mice groups used in the study.

Strain	Genotype	Phenotype	Mice Infection
SC5314	Wild type	Wild type	Group 1 Balb/c (N = 5)
DAY185	URA Reintegrated	Parental	Group 2 Balb/c (N = 5)
ES1	*dfg5:dfg5::dcw1:DCW1*	*DFG5* knockout strain with one copy of *DCW1*	Group 3 Balb/c (N = 5)
ES195	*dfg5:dfg5::dcw1:dcw1::MET3-DFG5*	No *DFG5* repression/2 mM methionine and 5 mM cysteine not added	Group 4 Balb/c (N = 5)
ES195+M/C	*dfg5:dfg5::dcw1:dcw1::MET3-DFG5* + 2 mM Methionine and 5 mM Cysteine	Conditional *DFG5*/*DCW1* mutant/5 mM methionine and 2 mM cysteine added for 1 h to medium for 85% *DFG5* repression	Group 5 Balb/c (N = 5) Mice were provided drinking water with 5 mM methionine and 2 mM cysteine from day 0 to day 5
HOG1	*hog1:hog1*	Hog1 knockout	None

Note: Genotypes of DFG5/DCW1 strains have been described previously by Spreghini et al., 2003 [12].

**Table 2 jof-10-00525-t002:** Primer sequences used in the study for qPCR analysis in *Candida albicans*.

Primers	Sequences (5′ TO 3′)	Tm	Source
*EFB1-F*	ATTGAACGAATTCTTGGCTGAC	63.0 °C	Munro et al., 2003 [23]
*EFB1-R*	CATCTTCTTCAACAGCAGCTTG	63.0 °C	
*CHS1-F*	GACAGTGGCAGTGACGATG	63.5 °C	Munro et al., 2003 [23]
*CHS1-R*	CAGCTTTGAGGTTGCTGC	62.3 °C	
*CHS2-F*	GGGAAAGATTCATGGAAGAAAATTG	62.0 °C	Kaneko et al., 2010 [24]
*CHS2-R*	TGCTTGTGCTCTTTCATTAATCTTTG	63.7 °C	
*CHS3-F*	TACGCTACTCCACCACATCAA	64.0 °C	Munro et al., 2003 [23]
*CHS3-R*	AAGAATACAAGAAATCAACCCTA	58.8 °C	
*CHS8-F*	GCCTTGTCTCCTTTACAACC	61.6 °C	Munro et al., 2003 [23]
*CHS8-R*	CTTGATGGTGGTACCACGTC	63.3 °C	
*CST20-F*	CACCAAGAACACCAACATCC	62.1 °C	This study
*CST20-R*	GACACACTCATGGAAGAAAGC	62.1 °C	
*HST7-F*	GCCAGCATTATCAAAATAGCCA	62.5 °C	qPrimerDB (ID#71336)
*HST7-R*	GTAAGATTTTCAGCACCGATCC	62.3 °C	
*CPH1-F*	TATGACGCTTCTGGGTTTCC	62.9 °C	This study
*CPH1-R*	GTGGAATCATGCCAATCATAGC	62.8 °C	
*CPH2-F*	GATTAGCAAAGTGGATGGTGTC	62.3 °C	qPrimerDB (ID#KHC73180)
*CPH2-R*	CACATGATTTTGTCCGTCAACT	62.4 °C	
*TEC1-F*	TCACCTTATGCTCAATATGGCA	62.8 °C	qPrimerDB (ID#KHC78996)
*TEC1-R*	GTGTTGGCTATTATGCGTGTAG	62.3 °C	
*EFG1-F*	ACAATGCAACAACCAACTCC	62.3 °C	This study
*EFG1-R*	TGTTACTCGTGGTCTGATTCC	62.4 °C	
*RIM101-F*	ATTGAAGCCTTTCCATTGTGAC	62.6 °C	qPrimerDB (ID#KHC841161)
*RIM101-R*	TAGTTGCATTCATCGAGTTTGC	62.5 °C	
*TUP1-F*	TAGACATTGCCAAAGCCAACC	64.3 °C	This study
*TUP1-R*	CAACTGACGAGTGGTCTAAGG	63.0 °C	
*RBF1-F*	CGACAAAGAATTGCTTACACCA	62.4 °C	qPrimerDB (ID#KHC73426)
*RFB1-R*	CAGGTGCATGATTATGTTGAGG	62.4 °C	
*RFG1-F*	GGTGGTGGTAGTATATCAGGTG	62.5 °C	qPrimerDB (ID#KHC71224)
*RFG1-R*	CTGTTGCTGTTGTTGTTGTAGT	62.5 °C	
*MIG1-F*	GCTTGTACATTTCCAGGTTGTG	63.0 °C	This study
*MIG1-R*	CCGTTTCCTTGAACTTGGATTG	63.0 °C	
*NRG1-F*	GTCGTCAAACAATAACACCCAA	62.4 °C	qPrimerDB (ID#KHC72092)
*NRG1-R*	ATTATCTTGACGAGCAAAACGG	62.3 °C	

## Data Availability

The original contributions presented in the study are included in the article, further inquiries can be directed to the corresponding author.

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
