# Peer review of "Candida albicans Mannosidases, Dfg5 and Dcw1, Are Required for Cell Wall Integrity and Pathogenesis"

_jof, 2024, doi:10.3390/jof10080525_

Round 1

Reviewer 1 Report

INTRODUCTION

1) The objective is poorly described

2) The role of the genes studied and their possible relationship with chitin synthesis must be justified, in the introduction and background section.

MATERIALS AND METHODS

3) Table 1 is incomplete, Candida albicans is never mentioned as the species of the strains. Information is missing in Table 1. Make a description of the strains as done in the article by Mancuso et al., 2018.

4) Authors can also write an extra column with the corresponding reference of the strains used

5) 2.4 Quantitative qRT-PCR the correct is RT-qPCR

The annealing temperature cannot be 95oC, it must be checked.

6) Table 2 could include the Tm of each of the oligonucleotides.

7) The words methionine and cysteine ​​are written with lowercase letters.

8) 2.5 Mouse Model

Make a Table that clearly describes the groups of mice, strains with which they were infected, controls, prior treatment of the strains (methionine and cysteine) and administration of methionine and cysteine ​​in the drinking water. By the way, it does not say at what concentration the amino acids were administered in the water.

RESULTS

9) The title of this result:

“3.1. “DFG5 and DCW1 Mutations Result in a Cell Separation Defect Identical to hog1 mutant”

It had already been demonstrated in the work of Mancuso et al., 2018, which is mentioned in the introduction of this manuscript. Therefore, it cannot be an original result.

10) “Fluorescence imaging using CFW that binds chitin revealed higher intensity of fluorescence at the cell septate, indicating increased chitin accumulation (Figure 1).”

Mention the panel of Figure: Fig. 1A

Is not clear, in which of the conditions or mutants is the higher intensity of Fluorescence observed?

11) “3.3. DFG5/DCW1 Mutations Result in Increased Chitin Levels

CFW fluorescence for ES195+M/C conditional mutant and hog1 knock-out mutant appeared to be increased as compared to WT, DAY185, and ES1 and ES195 strains (Figure 1).

This should be demonstrated with a statistical analysis. How many yeasts were observed? Was there a significant difference with the other strains or conditions?

12) “This is suggestive of increased chitin levels in the ES195+M/C conditional mutant and hog1 knock-out mutant, although the levels were much higher for the former. “normal chitin levels were observed for the ES1 mutant as measured by CFW fluorescence”

To be able to confirm the above, the authors should quantify chitin. If they do not want to use biochemical methods, may be, Flow Cytometry could be used.

13) In the Figure captions and Tables, Candida albicans should be mentioned.

14) The bottom of Fig. 4 lacks information. The yeast C. albicans is not mentioned. What was the route and site of inoculation, and how many CFU/mL were used to inoculate the mice?

15) This experiment in Fig 4 lacks a control: the mice without inoculating and making the same determinations, the histology and the viable count of the yeasts (even if there are no yeasts), immunosuppressing them, etc.

DISCUSSION

16) “This is the first study that describes novel upstream functions of

“Dfg5 and Dcw1 in negative transcriptional regulation of hyphal morphogenesis”

The results presented do not describe the above; this work's results only suggest this with indirect analyses.

REFERENCES

The references are not in the style requested by the Journal of Fungi. Many italic letters are missing from the scientific names of microorganisms. Sometimes, only one author is written in the reference, followed by et al., etc.

 In the Figure captions and Tables, Candida albicans should be mentioned.

REFERENCES

The references are not in the style requested by the Journal of Fungi. Many italic letters are missing from the scientific names of microorganisms. Sometimes, only one author is written in the reference, followed by et al., etc.

Author Response

I would like to sincerely thank the reviewer for their time and effort in reviewing our manuscript. Please see my responses to the comments below as a one by one basis. I am attaching two versions of the manuscript :

1) Manuscript with Track changes (in red)

2) Manuscript with accepted changes. 

Reviewer 1:

INTRODUCTION

Comment1: The objective is poorly described

Response:  Changes have been made to the last paragraph from the Introduction to describe the objective.

Comment 2: The role of the genes studied and their possible relationship with chitin synthesis must be justified, in the introduction and background section.

Response:  A paragraph has been added in the Introduction to describe the importance of chitin and genes related to chitin synthesis.

MATERIALS AND METHODS

Comment 3: Table 1 is incomplete, Candida albicans is never mentioned as the species of the strains. Information is missing in Table 1. Make a description of the strains as done in the article by Mancuso et al., 2018.

Response:   Table 1 has now been modified to include the genotype and phenotype descriptions. Additionally Candida albicans has been mentioned with reference to strains in the title for Table 1 as well as in the Methods section.

Comment 4: Authors can also write an extra column with the corresponding reference of the strains.

Response:  The reference for the strains is mentioned in the text of the Methods section.

Comment 5: Quantitative qRT-PCR the correct is RT-qPCR. The annealing temperature cannot be 95oC, it must be checked.

Response:  qRT-PCR has been corrected to “RT-qPCR” through out the text.

Comment 6: Table 2 could include the Tm of each of the oligonucleotides.

Response:  Table 2 has been modified to include Tm of the primers.

Comment 7: The words methionine and cysteine ​​are written with lowercase letters.

Response:  The words methionine and cysteine have been corrected to lowercase letters throughout the text.

Comment 8: 2.5 Mouse Model

Make a Table that clearly describes the groups of mice, strains with which they were infected, controls, prior treatment of the strains (methionine and cysteine) and administration of methionine and cysteine ​​in the drinking water. By the way, it does not say at what concentration the amino acids were administered in the water.

Response: Instead of making a new Table, we have added a column to Table 1 depicting the mice strains and groups corresponding to the respective Candida albicans strains. Additionally, Figure 1 has been added to depict the timeline of infection.

RESULTS

Comment 9: The title of this result:

  “3.1. “DFG5 and DCW1 Mutations Result in a Cell Separation Defect Identical to hog1 mutant”

It had already been demonstrated in the work of Mancuso et al., 2018, which is mentioned in the introduction of this manuscript. Therefore, it cannot be an original result.

Response: This has been corrected to “DFG5 and DCW1 Mutations Result in a Cell Separation Defect Identical to hog1 Mutant Confirmed by Light, Fluorescence and Electron Microscopy”

Comment 10: “Fluorescence imaging using CFW that binds chitin revealed higher intensity of fluorescence at the cell septate, indicating increased chitin accumulation (Figure 1).”

Mention the panel of Figure: Fig. 1A

Is not clear, in which of the conditions or mutants is the higher intensity of Fluorescence observed?

Response: The correction has been made to indicate Figure 2A instead of Figure 1.

Comment 11:  “3.3. DFG5/DCW1 Mutations Result in Increased Chitin Levels

CFW fluorescence for ES195+M/C conditional mutant and hog1 knock-out mutant appeared to be increased as compared to WT, DAY185, and ES1 and ES195 strains (Figure 1).

This should be demonstrated with a statistical analysis. How many yeasts were observed? Was there a significant difference with the other strains or conditions?

Response: CTCF measurements were performed for 100 cells/strain.  Paired T-test was performed between WT and mutant strains. There was significantly higher CTCF for ES195+M/C and Hog1 as compared to WT (SC5314) strain (p<0.05). Additionally, significantly higher CTCF was observed for ES195+M/C as compared to Hog1 (p<0.01). Data shown below.

T-Test paired

p-Value

WT Vs ES1

0.02484036

WT Vs ES195

0.16305896

WT Vs ES195+M/C

3.1882E-08

WT Vs Hog1

0.00767419

Hog1 Vs ES195+M/C

0.00025938

Comment 12:  “This is suggestive of increased chitin levels in the ES195+M/C conditional mutant and hog1 knock-out mutant, although the levels were much higher for the former. “normal chitin levels were observed for the ES1 mutant as measured by CFW fluorescence”

To be able to confirm the above, the authors should quantify chitin. If they do not want to use biochemical methods, may be, Flow Cytometry could be used.

Response: CFW fluorescence using CTCF measurements has been successfully used as an alternative approach for measuring chitin levels in C. albicans. Please see below references published in leading journals. We have added these references to the Methods section.

1. Walker LA, Munro CA, de Bruijn I, Lenardon MD, McKinnon A, Gow NA. Stimulation of chitin synthesis rescues Candida albicans from echinocandins. PLoS Pathog. 2008 Apr 4;4(4):e1000040. doi: 10.1371/journal.ppat.1000040. PMID: 18389063; PMCID: PMC2271054.

2. Liesche J et al., 2015. Cell wall staining with Trypan blue enables quantitative analysis of morphological changes in yeast cells. Microbiol. 6:107.

Comment 13: In the Figure captions and Tables, Candida albicans should be mentioned.

Response: Candida albicans has now been mentioned in all Tables and Figure captions.

Comment 14: The bottom of Fig. 4 lacks information. The yeast C. albicans is not mentioned. What was the route and site of inoculation, and how many CFU/mL were used to inoculate the mice?

Response: Inoculation was performed on day 0 with 106 cells/ml of C. albicans control and mutant strains in a calcium alginate oral swab for 75 minutes. This has been added to the text as well as Figure 1 and Figure 5 (previously Figure 4) caption.

Comment 15: This experiment in Fig 4 lacks a control: the mice without inoculating and making the same determinations, the histology and the viable count of the yeasts (even if there are no yeasts), immunosuppressing them, etc.

Response: A no infection group of mice was not included to minimize use of animals.

DISCUSSION

Comment 16: “This is the first study that describes novel upstream functions of

“Dfg5 and Dcw1 in negative transcriptional regulation of hyphal morphogenesis”

The results presented do not describe the above; this work's results only suggest this with indirect analyses.

Response: This sentence has been changed to “This is the first study that describes putative novel upstream functions of Dfg5 and Dcw1 in negative transcriptional regulation of hyphal morphogenesis”

REFERENCES

Comment 17: The references are not in the style requested by the Journal of Fungi. Many italic letters are missing from the scientific names of microorganisms. Sometimes, only one author is written in the reference, followed by et al., etc.

Response: The references have been formatted to meet the journal guidelines. The italics for scientific names have been corrected.

Detail comments

 Comment 18: In the Figure captions and Tables, Candida albicans should be mentioned.

Response: All Tables and Figure captions have a mention of C. albicans.

Reviewer 2 Report

This study builds upon previous publications by the authors. Because the fungal cell wall is a major target for the development of new antifungal drugs, this study is important.

There are 2 experiments that should be done to improve this manuscript:

1. Spot dilution assays for major cell wall stressors

2. Quantification of % cell separation for strains used in this study

Additional information is needed for the introduction and methods section

How did the authors define cell separation defect? While the cells seem to clump together, this could also be caused by increased cell-cell adhesion. And there are single cells in Figure 1A. It would be important to quantify the cell separation phenotype for control and chitinase treatment. Figure 1B does not have SEM images for the Hog1 mutant.

Another important experiment would be to subject strains to cell wall stressors and do spot-dilution assays. Results from these experiments could provide information on the underlying pathways.

Please expand the last paragraph of the introduction to include what was done, major take home messages and overall conclusions.

The original strain DAY185 has three auxotrophies. After re-integrating URA3 this strain is still His-, Arg-. Therefore I would not call this wild type parental and also would add the correct genotype for this strain, especially if it was used to generate additional mutants.

Please provide a reference for the methods, lines 140-144

Table 1 needs references for where the strains came from (from which original publication)

How did the authors define cell separation defect? While the cells seem to clump together, this could also be caused by increased cell-cell adhesion. And there are single cells in Figure 1A. It would be important to quantify the cell separation phenotype for control and chitinase treatment. Figure 1B does not have SEM images for the Hog1 mutant.

Another important experiment would be to subject strains to cell wall stressors and do spot-dilution assays. Results from these experiments could provide information on the underlying pathways.

It would highly recommend to not use phrases such as “Figure XYZ depicts….” It would be better to specifically talk about the results and then put the associated figure reference in parentheses.

It would also be helpful to the reader if each result section could state why this particular experiment was done (e.g., what was the question the authors wanted to answer).

Unless the reader is very familiar with the different pathways in Figure 5, the figure legend needs to provide a bit more information. Frankly, I don’t think this figure is really needed.

Please add a method section on statistical analysis and provide sufficient information on the types of t-tests etc.

The image resolution of Figure 3 is poor and needs to be improved.

I don’t feel the need to show the mouse tongue pictures in Figure 4 (or show only day1 and day5). The CFUs and histology results are sufficient.

Author Response

I would like to sincerely thank the reviewer for their time and effort in reviewing our manuscript. Please see my responses to the comments below as a one by one basis. I am attaching two versions of the manuscript :

1) Manuscript with Track changes (in red)

2) Manuscript with accepted changes. 

Reviewer 2:

Comment 1: This study builds upon previous publications by the authors. Because the fungal cell wall is a major target for the development of new antifungal drugs, this study is important.

Response: We thank the reviewer for their assessment.

Comment 2: There are 2 experiments that should be done to improve this manuscript:

  1. Spot dilution assays for major cell wall stressors

Response:  Spot dilution assays have been performed for ES1, E195 and ES195+M/C have been previously performed. Please refer to Mancuso et al., 2018.

  1. Quantification of % cell separation for strains used in this study

Response:  We have attempted that experiment but due to the clumping of cells in a random manner, the % cell separation was difficult to determine.

Additional information is needed for the introduction and methods section

Response: The Introduction and Methods section has been improvised. Tables 1 & 2 have been improved. Figure 1 has been added to describe the methods for the animal study.

Detail comments

Comment 3: How did the authors define cell separation defect? While the cells seem to clump together, this could also be caused by increased cell-cell adhesion. And there are single cells in Figure 1A. It would be important to quantify the cell separation phenotype for control and chitinase treatment. Figure 1B does not have SEM images for the Hog1 mutant.

Response: We have attempted that experiment but due to the clumping of cells in a random manner, the % cell separation was difficult to determine.

Comment 4: Another important experiment would be to subject strains to cell wall stressors and do spot-dilution assays. Results from these experiments could provide information on the underlying pathways.

Response: Spot dilution assays have been performed for ES1, E195 and ES195+M/C. Please refer to Mancuso et al., 2018.

Comment 5: Please expand the last paragraph of the introduction to include what was done, major take home messages and overall conclusions.

Response: We have now modified the Introduction to include what was done in the study. The major take home message and overall conclusions have been included in the Discussion and Conclusions sections respectively.

Comment 6: The original strain DAY185 has three auxotrophies. After re-integrating URA3 this strain is still His-, Arg-. Therefore, I would not call this wild type parental and also would add the correct genotype for this strain, especially if it was used to generate additional mutants.

Response: DAY185 has now been mentioned as Parental Control strain instead of Wild Type.

Comment 7: Please provide a reference for the methods, lines 140-144.

Response: The reference is Solis & Filler, 2012 and has now been added.

Comment 8: Table 1 needs references for where the strains came from (from which original publication)

Response:  Reference for the strains has been added in the Table title and also in the Methods section.

Comment 9: Another important experiment would be to subject strains to cell wall stressors and do spot-dilution assays. Results from these experiments could provide information on the underlying pathways.

Response:  Please see response above.

Comment 10: It would highly recommend to not use phrases such as “Figure XYZ depicts….” It would be better to specifically talk about the results and then put the associated figure reference in parentheses. It would also be helpful to the reader if each result section could state why this particular experiment was done (e.g., what was the question the authors wanted to answer).

Response: Changes have been made to the Results section to better describe the results.

Comment 11: Unless the reader is very familiar with the different pathways in Figure 5, the figure legend needs to provide a bit more information. Frankly, I don’t think this figure is really needed.

Response: We would like to include Figure 6 (previously Figure 5) as it graphically explains the location of Dfg5 and Dcw1 in the cell wall and it’s putative interactions with signaling proteins that are involved in MAPK pathways regulating cell wall integrity and hyphal morphogenesis.

Comment 12: Please add a method section on statistical analysis and provide sufficient information on the types of t-tests etc.

Response: Statistical analysis has been added to the Methods sections. The type of t-tests have been described.

Comment 13: The image resolution of Figure 3 is poor and needs to be improved.

Response: All images including Figure 3 have been updated to a Dpi of 300 for better resolution.

Comment 14: I don’t feel the need to show the mouse tongue pictures in Figure 4 (or show only day1 and day5). The CFUs and histology results are sufficient.

Response: Inclusion of the Figure 5 (previously Figure 4) for animal infections and tongue model allows a visual of development of the oral candidiasis. This is our first study using the oral candidiasis model and serves as a reference for us as well as the readers for following the development of infection over the five day period. We demonstrate in our study that we have followed the protocol in the most ideal manner to achieve our outcomes.

Reviewer 3 Report

The authors reported that the Candida albicans mannosidases Dfg5 and Dcw1 are required for pathogenesis. The entire manuscript is pretty preliminary and more details are needed to make the story convincing.

The introduction on these two enzymes and their role in fungi is rather meagre, important references on previous reports on these enzymes have not been discussed or citations are missing. The authors have to stick to results which are significant and based on clear statistics.

The materials and methods section is fragmentary and does not allow verification of the study by other scientists.

Furthermore, figure 1 is an over-interpretation of the results. The accumulation of chitin at the septae can only be seen for ES195+M/C and HOG1. However, the effect of the treatment with chitinase is not significant in Figure 1B. The box plots in figure 1C look strange and need more details. How many samples have been used for these calculations? The statement in the legend that HOG1 showed increased chitin accumulation has no substance according to figure 1C.

Figures 2 and 3 some error bars are missing, e.g. for DAY185

Introduction: It is important to state that C. albicans is not so much a pathogen but an opportunistic pathogen.

Line 63: delete one confirmed in this sentence.

A paragraph on statistics is missing. How many times have the experiments been repeated? S. D.s are based on how many samples?

The notation of the references has to follow the template of the journal, please correct.

Author Response

I would like to sincerely thank the reviewer for their time and effort in reviewing our manuscript. Please see my responses to the comments below as a one by one basis. I am attaching two versions of the manuscript :

1) Manuscript with Track changes (in red)

2) Manuscript with accepted changes. 

Reviewer 3:

Major comments

Comment 1: The authors reported that the Candida albicans mannosidases Dfg5 and Dcw1 are required for pathogenesis. The entire manuscript is pretty preliminary and more details are needed to make the story convincing.

Response: Major changes have been made to the manuscript beginning with Introduction and followed by Methods, Results and Discussion to make the story more convincing.

Comment 2: The introduction on these two enzymes and their role in fungi is rather meagre, important references on previous reports on these enzymes have not been discussed or citations are missing. The authors have to stick to results which are significant and based on clear statistics.

Response:  Several paragraphs have been added to the Introduction related to Dfg5 and Dcw1 as well as other genes being studied. A statistics section has been added to the Methods and the description of the statistical analysis of the experiments has been improvised for clarity on significance of results.

Comment 2: The materials and methods section is fragmentary and does not allow verification of the study by other scientists.

Response: Methods section has been modified to include more information in relation to Strains, CTCF calculations, animal study etc and also Table 1 has been improvised along with the addition of Figure 1 for animal study time line.

Detail comments

Comment 3: Furthermore, figure 1 is an over-interpretation of the results. The accumulation of chitin at the septae can only be seen for ES195+M/C and HOG1. However, the effect of the treatment with chitinase is not significant in Figure 1B.

Response: The results section has been modified to include the sentence that “accumulation of chitin is observed mainly for ES195+M/C and HOG1. The treatment of chitinase is significant in Figure 1B, however for SEM analysis the chitinase treated cells were settled on a glass slide coated with serum for analysis and were fixed using formaldehyde prior to imaging. Only one layer of cells could be viewed and that way and some of the cells appear to be not completely separated.

Comment 4: The box plots in figure 1C look strange and need more details. How many samples have been used for these calculations? The statement in the legend that HOG1 showed increased chitin accumulation has no substance according to figure 1C.

Response: Figure 1C has been updated now to reflect the statistically significant differences among strains. HOG1 CTCF levels were significantly higher than WT, hence our conclusion of increased accumulation of chitin is justified.

Comment 4: Figures 2 and 3 some error bars are missing, e.g. for DAY185

Response:  Error bars are not missing, the SD was very low for those experiments and so the error bars are not seen.

Comment 5: Introduction: It is important to state that C. albicans is not so much a pathogen but an opportunistic pathogen.

Response: This has been clarified in the Introduction.

Comment 6: Line 63: delete one confirmed in this sentence.

Response: This has been corrected in the Introduction.

Comment 7: A paragraph on statistics is missing. How many times have the experiments been repeated? S. D.s are based on how many samples?

Response: A paragraph on statistics has been added to the Methods section.

Comment 8: The light microscopy, fluorescence microscopy and SEM experiments were repeated a minimum of 2 times.

Response: For the CFW fluorescence experiments to determine chitin levels measurements were made for 50 cells/strain for two separate experiments. The measurements were them combined to obtain CTCF values for 100 cells/strain. The box plots are based on CTCF values for 100 cells/strain. For statistical analysis of significance, paired t-tests with unequal variances were performed with p-Value < 0.05. 

RT-qPCR experiments were performed for two separate experiments with triplicates of each strain/sample. The SD values are for a minimum of 6 samples per strain. For statistical analysis of significance, Student’s t-tests with equal variances were performed with p-Value < 0.05. 

Animal studies had 5 mice per group. CFU analysis was done in triplicate for each mouse. Student’s t-tests with equal variances were performed with p-Value < 0.05.  Histological analysis for every mouse. All of this information has been updated in the Methods section.

Comment 9: The notation of the references has to follow the template of the journal, please correct.

Response: The references have been correctly formatted according to Journal of Fungi requirements.

Reviewer 4 Report

Candida albicans mannosidases, Dfg5 and Dcw1, are required for pathogenesis

The authors have conducted a detailed study on the function of Dfg5 and Dcw1, showing their relationship with filamentation and pathogenesis. 

Introduction, Materials and Methods and Results are clear.

However, the discussion section must be improved.

In the first place, the sentences and cited references about resistance to caspofungin are not well ligated with the rest. 

In the second place, the hypothesis about the differences in the results of the 6h and 12h time points, although possible, is not supported by any experiment or reference. Besides, the reference by Lenardon (2017) is neither in the references section nor found in pubmed.  The cited reference from 2007 does not refer the experiments  the authors mention in the discussion section.

The authors should reform the discussion section before the manuscript is accepted. 

.abstract reiteration:

dfg5 and dcw1 mutations resulted in 

. line 63: repetition

. I have observed some bad use of medium or media and the subsequent verb. Please, revise. 

Author Response

I would like to sincerely thank the reviewer for their time and effort in reviewing our manuscript. Please see my responses to the comments below as a one by one basis. I am attaching two versions of the manuscript :

1) Manuscript with Track changes (in red)

2) Manuscript with accepted changes. 

Reviewer 4:

Candida albicans mannosidases, Dfg5 and Dcw1, are required for pathogenesis

Comment 1: The authors have conducted a detailed study on the function of Dfg5 and Dcw1, showing their relationship with filamentation and pathogenesis. Introduction, Materials and Methods and Results are clear.

Response: We thank the reviewer for their assessment

Comment 2: However, the discussion section must be improved.

Response: The discussion section has been improvised.

Comment 3: In the first place, the sentences and cited references about resistance to caspofungin are not well ligated with the rest. 

Response: We have tried our best to address this in the Discussion.

Comment 4: In the second place, the hypothesis about the differences in the results of the 6h and 12h time points, although possible, is not supported by any experiment or reference. Besides, the reference by Lenardon (2017) is neither in the references section nor found in pubmed.  The cited reference from 2007 does not refer the experiments the authors mention in the discussion section.The authors should reform the discussion section before the manuscript is accepted. 

Response: The differences in the 6h and 12h timepoint were an observation and we have only hypothesized a possible reason for such an effect in the Discussion. We were referring to the Lenardon 2007 study. This has been corrected in the Discussion section.

Detail comments

Comment 5: abstract reiteration:

Response: Abstract has been revised.

Comment 6: dfg5 and dcw1 mutations resulted in 

Response: This correction has been made in the abstract.

Comment 7: line 63: repetition

Response:  The repetition in the sentence has been corrected.

Comment 8: I have observed some bad use of medium or media and the subsequent verb. Please, revise. 

Response: This has been corrected in the text in the Methods section.

Round 2

Reviewer 1 Report

In general, the authors responded to the suggestions made to the manuscript of the first version.

In general, the authors responded to the suggestions made to the manuscript of the first version.

Author Response

Comment 1: In general, the authors responded to the suggestions made to the manuscript of the first version.

Response: We sincerely thank the reviewer for their time and effort in reviewing our manuscript.

Comment 2: In general, the authors responded to the suggestions made to the manuscript of the first version.

Response: We sincerely thank the reviewer for their assessment.

Reviewer 3 Report

The progress reported in this manuscript is still rather small and not impressing. A decent discussion with results from other studies would have elevated the level of this contribution a lot but the authors prefered to ignore that suggestion.

The statement of the authors that the format of the references has been corrected is not true. The citations of the references are still wrong. They have to be placed at the end of the sentence not at the beginning of the next. Again, stick to the format required by JoF.

Line 66: chitin synthases

Line 69: delete 4 at the beginning of the sentence

Author Response

Comment 1: The progress reported in this manuscript is still rather small and not impressing. A decent discussion with results from other studies would have elevated the level of this contribution a lot but the authors preferred to ignore that suggestion.

Response: We sincerely thank the reviewer for their time and effort in reviewing our manuscript.

Comment 2: The statement of the authors that the format of the references has been corrected is not true. The citations of the references are still wrong. They have to be placed at the end of the sentence not at the beginning of the next. Again, stick to the format required by JoF.

Response: The reference numbering has been corrected to be at the end of the sentence throughout the manuscript.

Comment 3: Line 66: chitin synthases

Response: "chitins" has been corrected to "chitin"

Comment 4: Line 69: delete 4 at the beginning of the sentence

Response: 4 has been deleted and moved to the end of the previous sentence.

Reviewer 4 Report

The manuscript has been improved but he authors and clearly describes the experiments and their relevance for Candida albicans pathogenesis and virulence. 

Please revise all the italics for Candida

Please correct the verb

It is noteworthy that there is no known upstream signaling pathways that has 417 

been identified for Tup1

Author Response

Comment 1: The manuscript has been improved but he authors and clearly describes the experiments and their relevance for Candida albicans pathogenesis and virulence. 

Response: We sincerely thank the reviewer for their time and effort in reviewing our manuscript.

Comment 2: Please revise all the italics for Candida

Response: This has been corrected throughout the text

Comment 3: Please correct the verb "acts" :line 417

Response: This has been corrected to "act"

Comment 4: It is noteworthy that there is no known upstream signaling pathways that 

been identified for Tup1

Response: Indeed which makes this paper very exciting in the field. Thank you.